# Toxic Gas and Smoke Generation and Flammability of Flame-Retardant Plywood

**DOI:** 10.3390/polym16040507

**Published:** 2024-02-13

**Authors:** Hee-Jun Park, Hao Jian, Mingyu Wen, Seok-Un Jo

**Affiliations:** 1Department of Housing Environmental Design, College of Human Ecology, Jeonbuk National University, Jeonju 54896, Republic of Korea; phjun@jbnu.ac.kr; 2Research Institute of Human Ecology, College of Human Ecology, Jeonbuk National University, Jeonju 54896, Republic of Korea; 3Wood Material Science and Engineering Key Laboratory, College of Material Science and Engineering, Beihua University, Jilin 132013, China; jianhao_1999@163.com

**Keywords:** flame-retardant plywood, flammability, toxic gas generation, smoke generation

## Abstract

Limited by flammability, wood and wood-based materials face challenges in distinguishing themselves as structural materials or finishing materials. Once burning, they can produce toxic gases detrimental to humans and the environment. Therefore, it is critical to make clear whether fire-retardant wood construction materials are insusceptible to fire and not the sources of toxic gases. This study aimed to evaluate flame-retardant plywood from the aspects of flammability and the toxic gas and smoke generation during combustion. The flame-retardant plywood was manufactured by impregnating a flame-retardant resin in line with International Maritime Organization (IMO) standards. The research results indicate that seven out of the eight kinds of toxic gases listed by the IMO, other than CO, were not detected during the combustion of the flame-retardant plywood. While CO was detected, its quantities under three test conditions are below the corresponding thresholds. Therefore, unlike synthetic resin products, flame-retardant plywood is a promising finishing material that can reduce the damage from toxic gases in the event of a fire. In the smoke generation tests, the mass reduction rate of flame-retardant plywood increased from 13% to 18% and then to 20% as the test condition became more severe. Under the same circumstances, the average maximum specific optical density also followed an upward trend, whose values (75.70, 81.00, and 191.20), however, still met the IMO standard of below 200. This reflects that the flame-retardant plywood is competent as a finishing material. Further, flammability was evaluated, and the critical flux at extinguishment (CFE), total heat release (Qt), and peak heat release rate (Qp) were determined to be 49.5 kW/m^2^, 0.21 MJ, and 0.66 kW, respectively, which all did not reach the corresponding thresholds given by the IMO. To sum up, flame-retardant plywood has satisfactory flame-retardant performance and meets fire safety standards, showing the potential to be an attractive finishing material for building and construction.

## 1. Introduction

Wood and wood-based materials have been widely used as structural and finishing materials for construction. However, their properties (constituents) render them vulnerable to fire [1].

In 2019, 40,103 cases of fire occurred in South Korea, which led to 2515 casualties and property damage of KRW 858.4 billion, as reported. Among the places where fires occurred, residential facilities ranked the top regarding frequency (11,058 cases, 27.6%), followed by industrial facilities (5429 cases, 13.5%). Among the initial igniting materials, paper, wood, and hay were the most common ones (9484 cases, 23.6%), followed by electricity-related materials and electronics (20.5%), garbage (11.3%), synthetic resins (11.2%), and food (7.9%). In addition, it was reported that about 75.8% of deaths (216 people) and about 79.7% of injuries (1777 people) occurred in residential and non-residential facilities [2]. Noteworthily, the number of large-scale fires is also on the rise every year. The soaring energy consumption and the utilization of miscellaneous combustible interior materials following rapid economic growth account for the increase in the number of fires and the aggravation of resulting damage, which are highly probable to intensify further in the future [3]. With inspiring technological and industrial advancements, various finishing materials have been developed and applied to buildings, which, however, can emit large amounts of toxic gases while burning. The most serious harm to humans in the event of a fire is asphyxiation after the inhalation of toxic gases rather than direct contact with flames [4]. The risk of such asphyxiation needs more vigilance nowadays as varieties of combustible synthetic polymer materials are gradually developed and used. Plenty of studies have been conducted on the types and characteristics of toxic gases generated in fires and the effects of these gases on the human body [3,5,6,7]. The International Maritime Organization (IMO) has determined eight kinds of toxic gases from fire that can cause fatal damage to the human body, namely carbon monoxide (CO), hydrogen bromide (HBr), hydrogen chloride (HCl), hydrogen cyanide (HCN), hydrogen fluoride (HF), nitric oxide (NO), nitrogen dioxide (NO2), and sulfur dioxide (SO2), for which thresholds have been set to facilitate strict management [8]. In addition, the IMO has provided flammability evaluation methods and safety standards for finishing materials and enacted relevant regulations so that only the products that conform to the standards can be selected for construction [9].

The following regulations were stipulated for the interior finishes and trims of buildings in the United States: interior finishing materials that give off smoke or gases denser or more toxic than those given off by untreated wood or untreated paper under comparable exposure to heat or flame shall not be permitted [10].

Besides being an important energy source, wood, along with its derived products, has also been used as a major component of buildings [7]. However, wood emits a variety of compounds when combusted [11]. By probing into the combustion behavior of flooring materials and the toxicity of generated gases, Lee et al. reported that the CO and CO_2_ emitted from wood-based medium-density fiberboard (MDF) flooring were less than those from polyester flooring and polyvinyl chloride (PVC) flooring [12]. In addition, they also evaluated the gas hazards from flame-retardant wood and the toxicity index of combustion gases and determined that the gas toxicity indexes of untreated wood and flame-retardant wood were 0.183 and 0.196–0.251, respectively, both lower than those of PVC (4.13) and urethane flooring (7.2) [13].

Wood and wood-based materials are sometimes restricted in application because they are vulnerable to fire. More seriously, fatal toxic gases are generated upon their combustion and pose threats to human health, which needs more consideration. Many efforts are made in research to lessen the vulnerability of wood and wood-based materials to combustion [1,14,15,16], and numerous valuable results have been achieved so far, including flame-retardant wood that does not burn at all. On this basis, it is necessary to study the types and amounts of the toxic gases generated when flame-retardant wood is combusted for a better understanding of wood and wood-based materials that are safer and can reduce harm to humans.

Phosphorus-based flame retardants attract more attention because of their high flame retardancy efficiency. On the one hand, they can capture free radicals in the gas phase; on the other hand, they promote the carbonization in condensed phase, which prevents the heat exchange and the release of pyrolysis volatiles. Therefore, phosphorus-based flame retardants are applied in various polymers extensively. Ammonium phosphate polymer (APP) and guanyl urea phosphate (GUP) are proven effective fire retardant chemicals. During combustion, by this intumescent fire retardant system, APP and GUP act as an acid and blowing agent, and wood materials provide carbon resources because of their ability to form a carbon layer when degraded.

This study was intended to evaluate the flammability and the toxic gas and smoke generation of flame-retardant plywood that is manufactured and applied as finishing materials for construction in South Korea, with combustion tests. The results are expected to contribute to the development of wood-based finishing materials with a higher safety level and reduced harm to humans upon combustion.

## 2. Experiment and Methods

### 2.1. Testing Materials

#### 2.1.1. Flame-Retardant Resin

A water-soluble, flame-retardant resin (NF200+, SAMHWA PAINTS Industrial Co., Ltd., Seoul, Republic of Korea) was used in the manufacturing of flame-retardant plywood. Its components mainly include ammonium phosphate polymer (APP), guanyl urea phosphate (GUP), phosphonic acid, acrylamide acrylic acid-N-(3-(dimethylamino)propyl)methacrylamide copolymer, 2-benzisothiazolin-3-one, and a minor amount of additives. The synthesized diagram is shown in Figure 1.

#### 2.1.2. Flame-Retardant Plywood

The flame-retardant plywood used for the evaluation of toxic gas and smoke generation and combustion characteristics was prepared through vacuum pressure impregnation with the flame-retardant resin (NF200+) under a pressure of 17 kgf/cm^2^ for 20 min. The impregnation amount of the flame retardant was more than 300 kg/m^3^, and the used flame-retardant plywood passed the flame-retardant performance test according to the KS F ISO 5660-1 standard [17]. The uses, specifications, and appearance before and after the test on the test specimens are shown in Table 1, Table 2, Table 3 and Table 4.

### 2.2. Test Methods

#### 2.2.1. Test Equipment

Toxic gas generation was measured by the Korea Marine Equipment Research Institute according to the test regulations of IMO Res. MSC. 307(88): 2010/ANNEX 1/Part 2. Figure 1 and Figure 2 show the block diagrams of the test equipment applied to the measurement of toxic gas generation and smoke generation, respectively.

#### 2.2.2. Test Methods

Toxic gas and smoke generation during the combustion of flame-retardant plywood was measured according to the test regulations of IMO Res. MSC. 307(88): 2010/ANNEX 1/Part 2 as shown in Figure 3. Before measurement, the test specimens were conditioned at a temperature of 23 ± 2 °C and relative humidity of 50 ± 5% for 816 h. The test methods and procedures for toxic gas and smoke generation upon combustion are shown in Table 2. In addition, the flammability of the flame-retardant plywood was measured in light of the test regulations of IMO Res. MSC. 307(88): 2010/ANNEX 1/Part 5. Before measurement, the test specimens were conditioned for 72 h under the same temperature and relative humidity conditions.

## 3. Results and Discussion

### 3.1. Toxic Gas Generation

In the event of a building fire, a wide variety of toxic gases are generated in the form of single gases or mixed gases depending on the combustion materials [4,6]. The IMO has determined eight kinds of toxic gases that may be generated when finishing materials for bulkheads, linings, or ceilings are combusted and set a criterion for each of them. Table 3 lists the eight kinds of toxic gases, their criteria, and their effects on the human body.

In this study, toxic gas and smoke generation from flame-retardant plywood upon combustion was measured according to the IMO fire safety standards for interior finishing materials (bulkhead, lining, and ceiling materials). Table 4 displays the appearance of the specimens before and after the 800 s tests.

Table 5 lists the emission results of the toxic gases from flame-retardant plywood upon combustion under three conditions (irradiance of 25 kW/m^2^ in the absence of a pilot flame, irradiance of 25 kW/m^2^ in the presence of a pilot flame, and irradiance of 50 kW/m^2^ in the absence of a pilot flame). Only CO was detected, while the others were not. In the three test conditions, 232 ppm, 293 ppm, and 1444 ppm CO were detected, lower than the threshold of 1450 ppm set by the IMO. Therefore, the flame-retardant plywood tested in this study was determined to be applicable as a finishing material for ships. In other cases, the detected toxic gases included 704 ppm CO, 663.6 ppm NO, 11 ppm SO_2_, 63 ppm HCl, and 70 ppm HF when polyurethane is combusted and 1830 ppm CO, 232.3 ppm NO, 7.0 ppm SO_2_, 282 ppm HCl, and 70 ppm HF when PVC is combusted [3]. Their distinct differences from the results of this study demonstrate that plywood made of wood is a safer material.

As is well known, wood is mainly constituted by elements of carbon (C, 50%), oxygen (O, 44%), and hydrogen (H, 6%). Consequently, only CO and CO_2_ are generated, while the other toxic gases are not released basically during wood combustion. In addition, the flame-retardant resin used in the manufacturing of the flame-retardant plywood in this study was soluble in water, whose main components were APP, GUP, phosphonic acid, acrylamide acrylic acid-N-(3-(dimethylamino)propyl)methacrylamide copolymer, 2-benzisothiazolin-3-one, and a small amount of additives [15]. In the combustion test of the flame-retardant plywood in which such a water-soluble, flame-retardant resin was introduced via vacuum pressure impregnation, no toxic gases other than CO among the eight kinds listed by the IMO were detected, and the amount of CO released was below the threshold. In other words, besides the gases generated from wood combustion, the emission of toxic gases due to the introduction of the flame-retardant resin also does not need excessive concern.

When selecting finishing materials, one should consider the above results to evaluate the effects of such materials on the environment and human bodies in case of a fire. Regarding wood and wood products, those that emit the smallest possible amount of toxic substances are good choices because people may suffer from minimal harm upon the combustion of these materials [7]. Flammability should not become the stumbling block in the application of wood. Since various types of flame-retardant wood, flame-retardant plywood, etc., are currently under development and production, it is expected that fire-retardant products have broad application prospects in providing buildings and residential environments that are safer from toxic gases generated by fire and can minimize the harm to humans once a fire occurs.

### 3.2. Smoke Generation

In this study, smoke generation during the combustion of flame-retardant plywood was measured according to the IMO fire safety standards for interior finishing materials (bulkhead, lining, and ceiling materials). Table 6 shows the average results of smoke generation from three test specimens measured in each condition.

The mass loss and maximum specific optical density (Dm) of the test specimens were evaluated under each test condition. The average mass of the test specimens was 87.48 g before the test and 72.30 g after the test, namely there was a mass decrease of 15.18 g on average under the test conditions. Therefore, the average mass reduction rate was calculated as 17%. Specifically, as the test conditions became more severe, the mass reduction rate tended to increase (from 13% to 18% and then to 20% in Figure 4).

The average Dm showed different values (75.70, 81.00, and 191.20) under the three test conditions. The IMO stipulated that the average Dm should not exceed 300 under these conditions. All the measurement results of average Dm in this study did not exceed 200 (Figure 5), indicating that the flame-retardant plywood meets the IMO smoke generation standard.

When flame-retardant plywood was applied as a finishing material, its Dm and toxic gas generation meet the finishing material standards presented by the IMO. Therefore, flame-retardant plywood can become a contributor to protecting humans from the considerable harm of toxic gases and smoke generated upon combustion.

### 3.3. Flammability

Table 7 lists the results of the flammability test on the flame-retardant plywood according to the standard under IMO Res. MSC. 307(88): 2010/ANNEX 1/Part 5. The test specimens were the flame-retardant plywood manufactured through vacuum pressure impregnation of the water-soluble flame-retardant resin mentioned in Section 3.1. The test was intended to examine the applicability of the foregoing material as a finishing material for ships.

As shown in Table 7, the critical flux at extinguishment (CFE) was 49.5 kW/m^2^ on average, which was more than twice the threshold of 20 kW/m^2^ given by the IMO. The average of the total heat release (Qt) was 0.21 MJ, which indicated excellent performance corresponding to about 30% of the upper threshold of 0.70 MJ presented by the IMO, and the peak heat release rate (Qp) was determined to be 0.66 kW on average, which was about 17% of the upper threshold of 4.00 kW given by the IMO, indicative of excellent flame-retardant performance. A previous study evaluated the flame-retardant performance of Korean pine as wall panels according to the test method under ISO 5660-1 [14]. The Korean pine was also subjected to the vacuum pressure impregnation of the same flame-retardant resin as that in this study. The evaluation results showed that the total heat release (THR) values of two samples were 6.24 MJ/m2 and 4.18 MJ/m2, which were below the upper threshold of 8.0 MJ/m2 and met the flame-retardant performance standard.

In conclusion, the test specimens in this study meet the requirements for low flame spread according to the surface flammability test procedures for surface materials specified in IMO Res. MSC. 307(88): 2010/ANNEX 1/Part 5.

By comparing the flame-retardant plywood shown in Figure 6, the flame-retardant plywood in this study showed excellent smoke suppression performance, with a CO output of only 1440 ppm, and a flame-retardant plywood with a high CO amount of more than 30,000. This study provides new methods and ideas for the research of flame retardant plywood [18,19,20].

## 4. Conclusions

Considering the flammability of wood and wood-based materials, people usually hesitate to select them as building structural materials or finishing materials. It has been reported that the inhalation of toxic gases generated during combustion is responsible for most of the enormous damage to humans in the event of a fire rather than direct contact with flames. In recent years, fire-safe flame-retardant wood has been developed after unremitting efforts. In this study, flame-retardant plywood manufactured with a flame-retardant resin impregnated in it was evaluated in terms of toxic gas and smoke generation upon combustion and flammability according to IMO standards, and the following conclusions were drawn.

(1)As for the toxic gases generated during the combustion of flame-retardant plywood, among the eight kinds of toxic gases presented by the IMO, seven kinds other than CO were not detected. Even if CO was detected, its amounts did not exceed the thresholds under three test conditions. Therefore, unlike synthetic resin products, flame-retardant plywood was evaluated as a finishing material that can reduce the damage from toxic gases in the event of a fire.(2)In the smoke generation tests, the mass reduction rate showed a tendency to increase (13%, 18%, and 20%) as the test condition became more severe. In addition, the average Dm got higher (75.70, 81.00, and 191.20) under a more harsh condition but still met the standard of below 200 given by the IMO. This manifests that flame-retardant plywood is applicable as a finishing material.(3)The flammability of flame-retardant plywood was evaluated, and the values of CFE, Qt, and Qp were determined as 49.5 kW/m^2^, 0.21 MJ, and 0.66 kW, respectively, which were all within the acceptable ranges set by the IMO. Therefore, the applicability of flame-retardant plywood as a finishing material meets the fire safety standards.(4)The flame retardant characterization analysis verified that fire retardant chemicals with the APP, GUP, and phosphonic acid as main component treatments effectively accelerated the dehydration and carbonization of wood. Consequently, more char and less flammable volatile products resulted, and a better fire retardant performance was obtained.(5)This study provides evidence for the fact that fire-retardant wood and wood-based materials can also possess the capability against fire. These materials are expected to create buildings and residential environments that are safer from toxic gases generated by fire and minimize damage to humans.

The major approach to enhancing the surface function of wood has increasingly shifted toward low-carbon, halogen-free, and non-toxic alteration techniques as a result of ongoing contemplation on human development and environmental issues. Future development will focus heavily on using biomass as a raw material.

They are seen to be potentially significant ways to alter the course of wood’s future. Enhancing the functionality of wood is still a hot topic in wood research in general. The researchers improved their approach and the data they compiled in the publication are available for research and reference. In the future, it is anticipated that new and better techniques will be developed to enhance the flame retardancy, hydrophobicity, and antibacterial properties of wood surfaces.

## Figures and Tables

**Figure 1 polymers-16-00507-f001:**
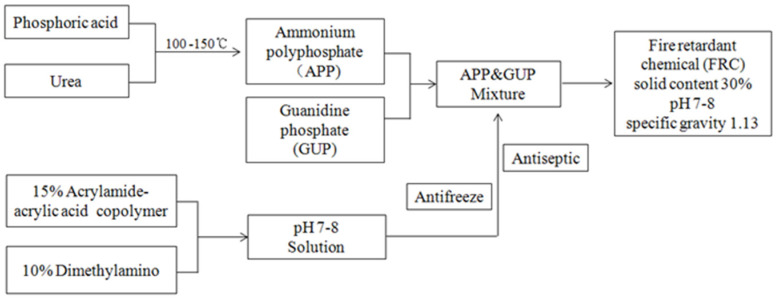
Flame retardant synthesis diagram.

**Figure 2 polymers-16-00507-f002:**
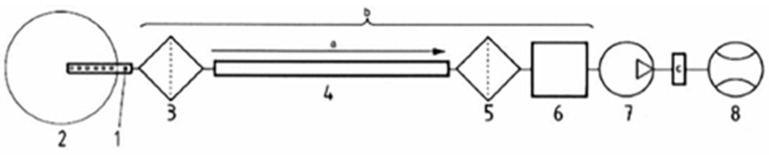
Test equipment arrangement for measuring toxic gas generation. 1. Probe, 2. Exhaust duct (fire model), 3. Filter 1, 4. Transfer line, 5. Filter 2, 6. Gas cell, 7. Pump, 8. Flow meter.

**Figure 3 polymers-16-00507-f003:**
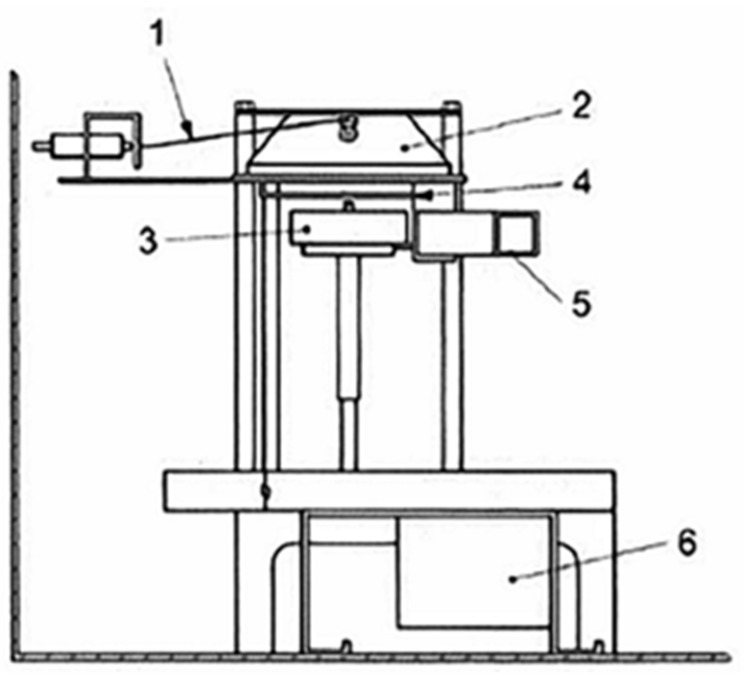
Test equipment arrangement for measuring smoke generation. 1. Thermocouple, 2. Radiator cone, 3. Specimen holder, 4. Radiation shield, 5. Heat flux meter holder, 6. Spark ignition housing.

**Figure 4 polymers-16-00507-f004:**
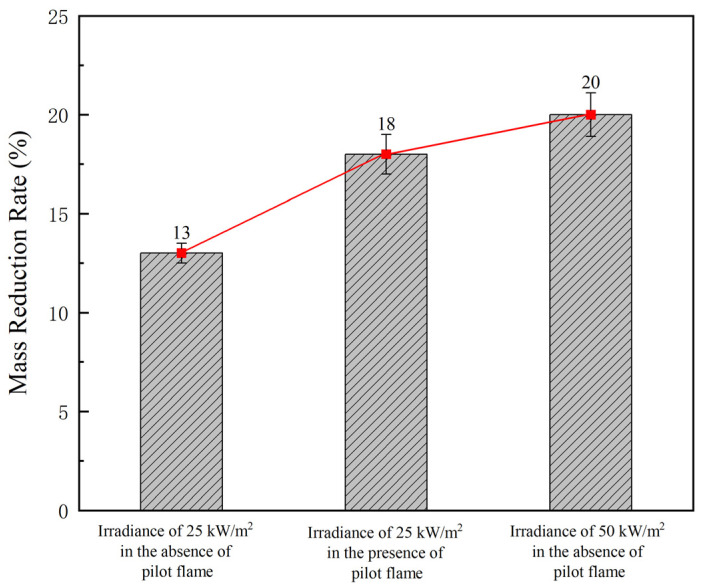
Mass reduction rate in each test condition.

**Figure 5 polymers-16-00507-f005:**
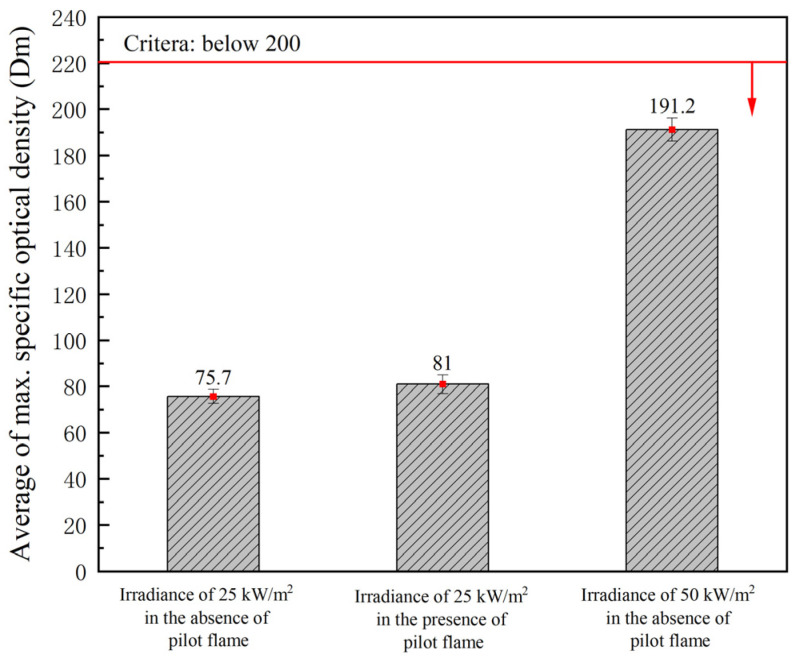
Average Dm in each test condition.

**Figure 6 polymers-16-00507-f006:**
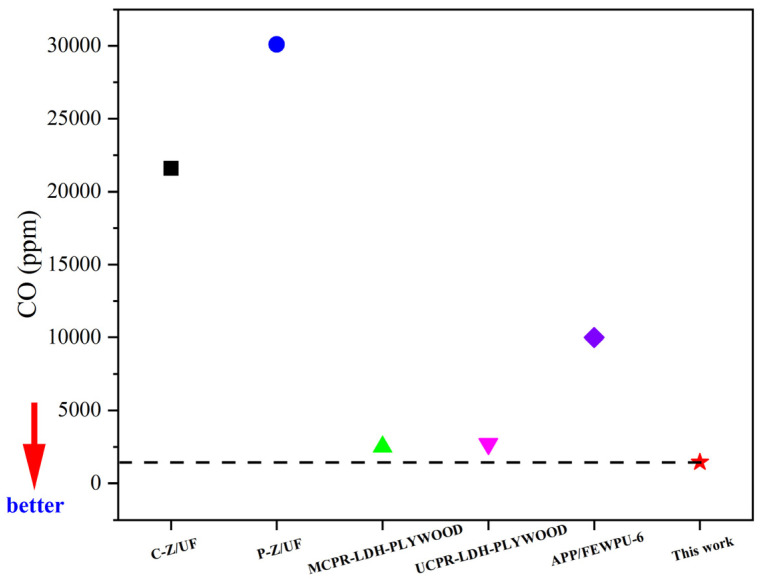
Comparison of flame-retardant plywood.

**Table 1 polymers-16-00507-t001:** Specifications of test specimens [8,9].

Construction	Uses	Toxic Gas and Smoke Generation		Flammability		Flame-Retardant Resin
		Dimensions (mm)	Number of Specimens	Dimensions (mm)	Number of Specimens	
Plywood	Bulkheads, Linings, Ceilings	W75 × L75 × T24	9	W155 × L800 × T24	3	NF200^+^ (SAMHWA PAINTS Industrial Co., Ltd.)

**Table 2 polymers-16-00507-t002:** Test methods of toxic gas and smoke generation upon combustion.

Procedure	Test Methods	
	Toxic Gas Generation	Smoke Generation
1	Remove all dirty layers and particles in the test chamber, and clean the internal probe.	Prepare the test chamber Prepare the test chamber with a cone set at 25 kW/m^2^ or 50 kW/m^2^, set the distance between the cone heater and the specimen to be 50 mm, and position the pilot burner 15 mm down from the bottom edge of the cone heater.
2	Maintain the filters, gas sampling line, valves, and gas cell at 150–180 °C for at least 10 min prior to test.	Tests with pilot flame For tests with a pilot flame, make the burner in position, turn on the gas and air supplies to ignite the burner, and check the flow rates.
3	During the smoke density test, start sampling by opening the sampling valve to introduce the gas in the chamber into the sampling line at the moment of maximum smoke density.	Preparation of photometric system Perform zero setting, open the shutter to set the full-scale 100% transmission reading, recheck the 100% setting, and repeat the operations until accurate zero and 100% readings are obtained on the amplifier and recorder when the shutters are opened and closed.
4	-	Loading the specimen Place the holder and specimen on the supporting framework below the radiator cone, remove the radiation shield from below the cone, and simultaneously start the data recording system and close the inlet vent. The test chamber door and the inlet vent must be closed immediately after the test starts.
5	-	Recording of light transmission Record the light transmission and time continuously from the start of the test.
6	-	Termination of test The initial test in each test condition must last for 20 min to verify the possible existence of a second minimum transmittance.
7	-	Conditioning of specimens Before measurement, the test specimens must be conditioned to a constant mass at 23 ± 2 °C and 50% ± 2% relative humidity.

**Table 3 polymers-16-00507-t003:** Toxic gases generated upon combustion and their characteristics and criteria [5,6,8].

Toxic Gas	Criterion (ppm)
CO	≤1450
HBr	≤600
HCl	≤600
HCN	≤140
HF	≤600
SO_2_	≤120
NO	≤350
NO_2_	≤350

**Table 4 polymers-16-00507-t004:** Appearance of test materials under different test conditions.

Test and Condition		Before the Test	After the Test
Toxic gas and smoke generation (IMO, Part 2)	Irradiance of 25 kW/m^2^ in the absence of pilot flame	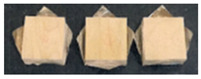	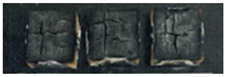
	Irradiance of 25 kW/m^2^ in the presence of pilot flame	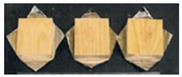	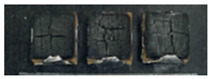
	Irradiance of 50 kW/m^2^ in the absence of pilot flame	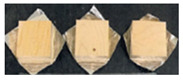	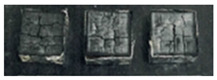
Flammability (IMO, Part 5)	Heat flux: 50.5 kW/m^2^ (at the 50 mm position) 23.9 kW/m^2^ (at the 350 mm position)	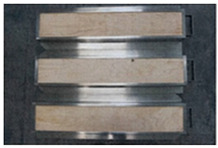	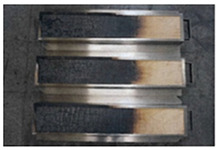

**Table 5 polymers-16-00507-t005:** Toxic gas generation of flame-retardant plywood upon combustion.

Toxic Gas	Irradiance of 25 kW/m^2^ in the Absence of Pilot Flame	Irradiance of 25 kW/m^2^ in the Presence of Pilot Flame	Irradiance of 50 kW/m^2^ in the Absence of Pilot Flame	Criterion (ppm)
CO	232	293	1444	1450
HBr	0	0	0	600
HCl	0	0	0	600
HCN	0	0	0	140
HF	0	0	0	600
SO_2_	0	0	0	120
NO	0	0	0	350
NO_2_	0	0	0	350

Notes: These criteria are set for bulkheads, linings, or ceilings. Each value is the average result of three tests in each condition.

**Table 6 polymers-16-00507-t006:** Smoke generation of flame-retardant plywood upon combustion.

Parameter	Irradiance of 25 kW/m^2^ in the Absence of Pilot Flame	Irradiance of 25 kW/m^2^ in the Presence of Pilot Flame	Irradiance of 50 kW/m^2^ in the Absence of Pilot Flame	Criterion
Initial mass (g) (±0.02)	85.92	86.70	89.82	Dm must not exceed 200 in any test condition
Final mass (g) (±0.02)	74.38	71.02	71.49	
Mass loss (g) (±0.02)	11.53	15.68	18.33	
Average Dm	75.70	81.00	191.20	

Notes: The criterion is set for bulkheads, linings, or ceilings. Each value is the average result of three tests in each condition.

**Table 7 polymers-16-00507-t007:** Flammability of flame-retardant plywood.

		Specimen 1	Specimen 2	Specimen 3	Average	Surface Flammability Criterion ^1^
Item						
Average heat for sustained burning, Qsb (MJ/m^2^)		-	-	-	-	≥1.50
Critical flux at extinguishment, CFE (kW/m^2^)		49.5	49.5	49.5	49.5	≥20.0
Total heat release, Qt (MJ)		0.30	0.16	0.17	0.21	≤0.70
Peak heat release rate, Qp (kW)		0.87	0.53	0.58	0.66	≤4.00
Burning droplets		None	None	None	None	Not produced

Notes: ^1^ These criteria are set for surface materials (bulkhead, wall, and ceiling linings). Since there was no ignition, no values were recorded.

## Data Availability

Data are contained within the article.

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
