# Peer review of "Toxic Gas and Smoke Generation and Flammability of Flame-Retardant Plywood"

_polymers, 2024, doi:10.3390/polym16040507_

Round 1

Reviewer 1 Report

Comments and Suggestions for Authors

In the introduction I would prefer to see the information concerning the results of the studies of the same or different retardant additives rather than the details on the number of fire incidents.

In the Experimental there are many exessive details. It is not a good choice to present the whole instruction of the equipment including closing the door and zero setting etc. (page 4).

It was a priory evident that bromine-containing gases and other harmful gases listed exept for CO would not appear from burning wood. Does it make any sense to describe this finding?

Is it necessary to remind the readers about the well-known  harm of the toxic gases (table 3, coloumn 2)?

What is really necessary - to describe the role (mechanism) of flame-retardant resin. In this case the manusript would correspond to the scope of the journal.

Author Response

#Question 1: In the introduction I would prefer to see the information concerning the results of the studies of the same or different retardant additives rather than the details on the number of fire incidents.

Lines 94-102, We have added the information concerning the results of the studies of the same retardant additives.

#Question 2: In the Experimental there are many exessive details. It is not a good choice to present the whole instruction of the equipment including closing the door and zero setting etc. (page 4).

We present a more complete experimental procedure designed to demonstrate the feasibility and accuracy of the experiment.We have revised some contents accordingly.

#Question 3: It was a priory evident that bromine-containing gases and other harmful gases listed exept for CO would not appear from burning wood. Does it make any sense to describe this finding?

Flame-retardant plywood contain adhesives and flame retardants in addition to nature wood, and adhesives and traditional flame retardants are prone to produce toxic gases, such as bromine gas, in addition to CO when burned.

#Question 4: Is it necessary to remind the readers about the well-known harm of the toxic gases (table 3, coloumn 2)?

We've streamlined and modified the redundancies.

#Question 5: What is really necessary - to describe the role (mechanism) of flame-retardant resin. In this case the manusript would correspond to the scope of the journal.

In the final conclusion section, we have added synthesis diagram of the flame retardants and explanation of the flame retardant mechanism.

Reviewer 2 Report

Comments and Suggestions for Authors

The manuscript "Toxic Gas and Smoke Generation and Flammability of Flame-Retardant Plywood" are well written but rather as a report than a scientific paper. The manuscript needs revision. The abstract, conclusion and introduction provide a good readable structure. There are some parts missing in the results.

1. The authors said they used FTIR but there no FTIR shown. please include the FTIR before and after burning of the different procedure. This can be placed in supplementary.

2. There are no real discussion provided in the result part to other work made in the field. This is the main drawback from the manuscript. Either include a separate section called discussion or add it to the result part.

3, From the specimen before and after burning shown in Table 4, what time of burning is applied and did the authors used longer time until only ashes obtained? 

4. It would be beneficial showing how those flame retardant used in the plywood in this work inhibit gas release. Either a formular, scheme or more in detail described text would help.

5. Additionally, a table of comparison to other work where plywood with flame retardants are used are as well helpful. Such can be added in the new section of discussion part

Author Response

#Question 1: The authors said they used FTIR but there no FTIR shown. please include the FTIR before and after burning of the different procedure. This can be placed in supplementary.

We have deleted the inappropriate parts.

#Question 2: There are no real discussion provided in the result part to other work made in the field. This is the main drawback from the manuscript. Either include a separate section called discussion or add it to the result part.

We have added discussion in conclusion section about the explanation of the mechanism of flame retardants and an outlook on the future application directions and research directions.

#Question 3: From the specimen before and after burning shown in Table 4, what time of burning is applied and did the authors used longer time until only ashes obtained?

The test time is 800 s, usually at 50 kW/m2 power after 800s the trend of total smoke release tends to be flat, so the test time is set at 800s.

#Question 4: It would be beneficial showing how those flame retardant used in the plywood in this work inhibit gas release. Either a formular, scheme or more in detail described text would help.

In the final conclusion section, we have added explanation of the flame retardant mechanism.

#Question 5: Additionally, a table of comparison to other work where plywood with flame retardants are used are as well helpful. Such can be added in the new section of discussion part

We reviewed similar studies on flame retardant plywood, and there are relatively few studies on the harmfulness of gases, and most smoke production is talking about yield of CO,CO2

Round 2

Reviewer 1 Report

Comments and Suggestions for Authors

The text is reconsidered totally. To my opinion, it is more appropriate for the Journal now.

Author Response

Thank you for your comments on our manuscript. We also thank you for recognizing our manuscript and providing valuable guidance to our researchers.

Reviewer 2 Report

Comments and Suggestions for Authors

The revised manuscript does not changed from the former one only that all text was before black is now in red. The authors did not add any discussion to other works. This must be included. please add that

Author Response

At your request, we have added the necessary comparative discussion with other works. Changes made in the original manuscript have been noted.